# Optical Interference Suppression Based on LCD-Filtering

**Gilbert J. M. Forkel *** , **Adrian Krohn and Peter A. Hoeher ***

Chair of Information and Coding Theory, University of Kiel, Kaiserstr. 2, 24143 Kiel, Germany
* Correspondence: gf@tf.uni-kiel.de (G.J.M.F.); ph@tf.uni-kiel.de (P.A.H.)

**Abstract:** Using light emitting diodes (LED) for the purpose of simultaneous communication and illumination is known as visible light communication (VLC). Interference by ambient light sources is among the most critical challenges. Owing to the wideband VLC spectrum, the efficiency of wavelength-dependent optical filtering is limited, especially in the presence of sunlight. Multi-user VLC causes additional interference, since LEDs are characterized by a wide viewing angle. Although algorithm-based interference suppression is a feasible method, receiver saturation and especially noise enhancement are two challenges that can only by addressed effectively by filtering in the optical domain prior to the photodetector. In this publication, we propose the use of a liquid-crystal display (LCD) as receiver-side filter unit. The main advantage of this technology is the possibility to focus the field-of-view of the receiver on a specific light source and thereby suppress interference. Interference by ambient light, modulated interference and multi-aperture interference are introduced and signal-to-interference ratio improvements are derived using experimental results for a given LCD characteristic. By deriving the bit error rate for MIMO communications, the potential of the proposed interference reduction method is demonstrated.

**Keywords:** optical wireless communication; visible light communication; free-space communication; liquid-crystal display; optical filtering; interference reduction; ambient light suppression; MIMO

## 1. Introduction

In the field of visible light communication (VLC), interference reduction is a key component towards the realization of reliable communication links [1] (Chapter 9). Unlike conventional radio frequency communication, so-called direct detection receivers using one or several photo diodes (PDs) exhibit a much broader spectral response than required for detecting the useful signal. As a consequence, additional filtering is required to reduce interference. Owing to the wide-angle beam pattern of LEDs and the strong signal power attenuation increasing with distance, interfering sources and especially the solar irradiation often outclass the useful signal.

Of course, it is possible to reduce the interference after detection by means of filtering in the electrical domain. In the simplest case, the interfering signal is of unmodulated nature as it can be expected for sunlight or contains only low frequency components such as the line frequency with its harmonics when electric lighting is present. In those cases, simple low-pass filtering can be applied while for stochastic interference signals algorithmic approaches such as successive interference cancellation (SIC) have to be used. However, this increases the computational complexity of the receiver, reduces the system performance and is sometimes unfeasible. Besides these limitations of electrical domain filtering, noise enhancement is inevitable since shot-noise is induced at the moment when the interfering signals are detected by the PD. Furthermore, receiver saturation due to high interference power levels has to be taken into account.

The most prominent method for interference reduction is wavelength-dependent filtering [2]. Different filter types like colored glass filters and dichroic filters are available, sharing the principle challenge of suppressing unwanted spectral components while preserving the communication signal as completely as possible. The major limitation of dichroic filters is the angular derivation of the filter response [3]. This is a major problem for communication systems where the angle of arrival of the optical signal is not known a priori. In the case of colored glass filters, the filter transitions are not as sharp, leading to impaired transmissivity values when narrow passbands are required to match the LED spectral characteristic. Naturally, spectral filtering does not allow for suppressing interference in the wavelength regime of the desired signal. This is a critical limitation in LED-based communication systems with relatively wide transmitter spectra, e.g., when compared to laser diodes.

Different filter architectures taking the direction of the incoming signals into account have been proposed. One approach consists of using a digital camera set-up [4,5], where the individual PD pixels can be mapped to the incident angle of the incoming light rays such that filtering by the direction of the light sources is possible. This makes it possible to reduce the interference by selecting the detector pixels receiving the useful signal while dropping all others. However, unlike conventional PD receivers, this method requires a huge amount of photo cells to achieve a reasonable signal separation. This unavoidably leads to a reduction of the size of the individual detectors with an associated performance penalty to the useful signal. Moreover, this kind of direction-dependent filtering requires comparatively large hardware resources since only a small fraction of the available resources is used for signal detection. Imaging receivers [6–8] mapping the incident light rays on multiple PDs using lenses do not show this limitation. Owing to the static nature of the filters, channel decorrelation relies on a suitable positioning and orientation of the transmitters and receivers. Similar arguments apply to non-imaging angle-diversity receivers like the mirror-based approach described in [9].

Another possibility is presented by the authors in [10]. Here, a small number of PDs are combined with a repetitive static aperture. For example, by using eight PDs, an almost perfect suppression of an interfering light source can be achieved with a probability of $15/16$ when assuming random distributed point light sources.

In this contribution, we introduce a novel filter approach using liquid-crystal display (LCD) technology, replacing the static aperture by a dynamic one and thereby expanding the range of application to a wider scope of communication scenarios. A related optical filtering method has been presented in [11], employing transmitter-side beam-stearing using an LCD. A significant difference to our approach is the reduction of the optical output power by transmitter-side filtering, impairing the illumination functionality of VLC. Furthermore, interference by other light sources like unmodulated lighting fixtures and by solar irradiation can only be reduced by receiver-side filtering.

The fundamentals of operation as well as the experimental set-up under investigation are introduced in Section 2, discussing the geometric arrangement and the relevant optical parameters. In Section 3, the LCD filter technique is evaluated with respect to the signal-to-interference ratio (SIR) improvement for two different interference scenarios. A MIMO scenario is introduced in Section 4 and simulation results on the bit error rate (BER) performance are presented. Finally, conclusions are drawn in Section 5.

## 2. Fundamentals and Experimental Set-Up

The dynamic filter approach presented can be considered as an analogon to receiver-side beamforming, well known in RF communication, allowing for steering the receive beam pattern. This is accomplished by mounting a transmissive LCD in front of a PD and controlling the individual LCD pixels in order to maximize the SNR. The arrangement under consideration is depicted in Figure 1. By switching the transmissivity of the individual pixels the LCD can be adjusted such that light from specific origins is able to pass the LCD while the interfering signals are mostly blocked. In the simplest case, the LCD filter is targeted at a single transmitter by setting one area of LCD pixels to transmissive state while all other pixels are blocking incident light rays.

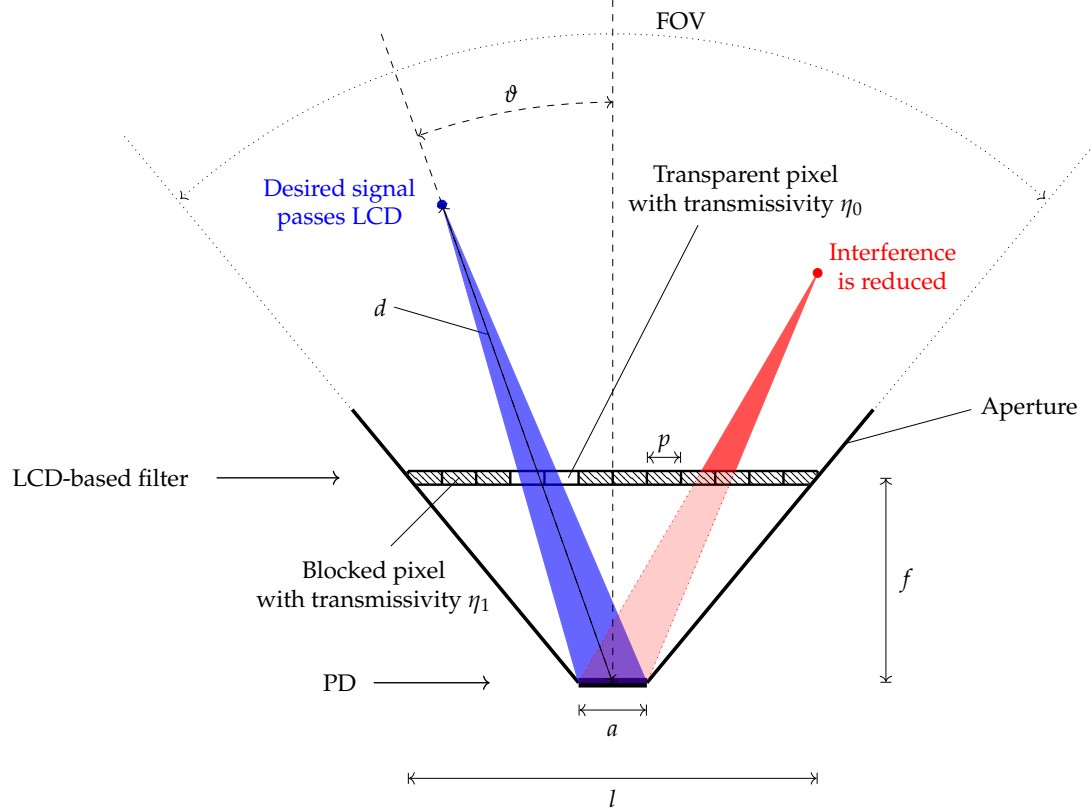

**Figure 1.** Geometry of detector with the LCD-based pre-filter.

## 2.1. Geometric Considerations

Let us first discuss the geometric requirements. Based on the two-dimensional sketch in Figure 1, we can derive the distance between LCD and PD as

$$f = \frac{l}{2 \cdot \tan\left(\text{FOV}/2\right)}, \tag{1}$$

with

$$l \gg a, \tag{2}$$

where $a$ is the edge length of the PD and $l$ the size of the LCD filter. For simplicity, we restrict to a PD with square detector area, resulting in an aperture similar to a pyramidal horn antenna.

Next, we consider a point light source at distance $d$ from the PD. A light beam emanating from this source and targeting to the PD is passing a maximum number of

$$N = \left\lceil \frac{a}{p}\left(1 - \frac{f}{\cos\vartheta \cdot d}\right) \right\rceil + 1 \tag{3}$$

LCD pixels per dimension. If we furthermore require $d \gg f$, we can interpret the incident light as a bundle of parallel rays with a width similar to the size of the PD. Correspondingly, the above result simplifies as

$$N \approx \left\lceil \frac{a}{p} \right\rceil + 1. \tag{4}$$

When switching all $N$ pixels to the transmissive mode, the angular width of the resulting beam pattern as a function of the angle of incident $\vartheta$ is given by

$$\Theta\left(\vartheta\right) = \arctan\left(\frac{N \cdot p}{2 \cdot f} + \tan\vartheta\right) + \arctan\left(\frac{N \cdot p}{2 \cdot f} - \tan\vartheta\right) \tag{5}$$

and can be later used for the calculation of the remaining interference passing the LCD at its opened section.

## 2.2. LCD Transmissivity

Using an LCD as an optical beamforming filter, we restrict ourselves to the two LCD pixel states "transmissive" and "opaque". For $s = 0$, the LCD pixels are transmissive with transmission coefficient $\eta_0(\lambda, \vartheta)$ while for $s = 1$ only a small fraction $\eta_1(\lambda, \vartheta)$ of the incoming light arrives at the detector. Thus, we can fully model the transmissivity as

$$\eta(\lambda, \vartheta) = \eta_0(\lambda, \vartheta) + s \cdot (\eta_1(\lambda, \vartheta) - \eta_0(\lambda, \vartheta)) . \tag{6}$$

The selected transmission coefficient depends on the wavelength $\lambda$ and the angle $\vartheta$ of the incoming light rays.

Generally speaking, LCDs are highly-optimized commercial products and are undergoing a continuous development owing to their commercial importance. Nowadays, almost all LCDs are offered as colored screens with integrated backlight. High-performance gray-scale screens are typically available as application-specific products only. Furthermore, datasheets usually specify the contrast ratio but not the transmissivity in transparent state, and thus we decided to base our research on measurements conducted on a large-area liquid-crystal (LC) cell.

These measurements were carried out by targeting a narrow light beam on the LC cell and measuring the received power for $s \in \{0, 1\}$ using a Si-PD of type FDS 100 from Thorlabs Inc. (Newton, NJ, USA), connected to a Keithley 2450 source-measurement unit (SMU). The LC cell is of size 60 mm $\times$ 90 mm and is advertised as a low-cost optical shutter e.g., used in auto-darkening welding helmets. As a light source, an LED Engin LZC-03MD07 module comprising four different LED colors with spectral distribution shown in Figure 3 is used. The measurement results in Figure 2 are obtained by normalizing the received powers for the LC cell states "on" and "off" by the received power without the LC cell in the optical path. Since the orientation of the LCD has a comparatively low impact on the measurement results, we specified the transmissivity values by averaging the measurement results in respect to the LC cell rotation. In order to enable an analytic evaluation, the LC transmissivity has been approximated for the 454 nm LED as

$$\eta_0(\vartheta) = 0.297 \cdot \cos(0.81 \cdot \vartheta) \tag{7}$$

and

$$\eta_1(\vartheta) = 1.0001 - \cos(0.071 \cdot \vartheta) . \tag{8}$$

As shown in Figure 2, this approximation is very close to the measurement results for incident angles $\vartheta \leq 60°$.

The measurements in Figure 3 are obtained using a BTS 256-EF spectrometer by Gigahertz-Optik (Türkenfeld, Germany). In case of the LC transmissivity, relative measurements are performed using sunlight. Notably, the wavelength-dependent transmissivity variation of the LC cell is moderate in the range from 400 nm to 750 nm. Below 400 nm, the LC cell strongly attenuates the receive signal, and above 750 nm the LC cell is transparent for the optical signals. To avoid infrared interference leaking into the LC cell, a combination with an infrared blocking filter is an apparent solution.

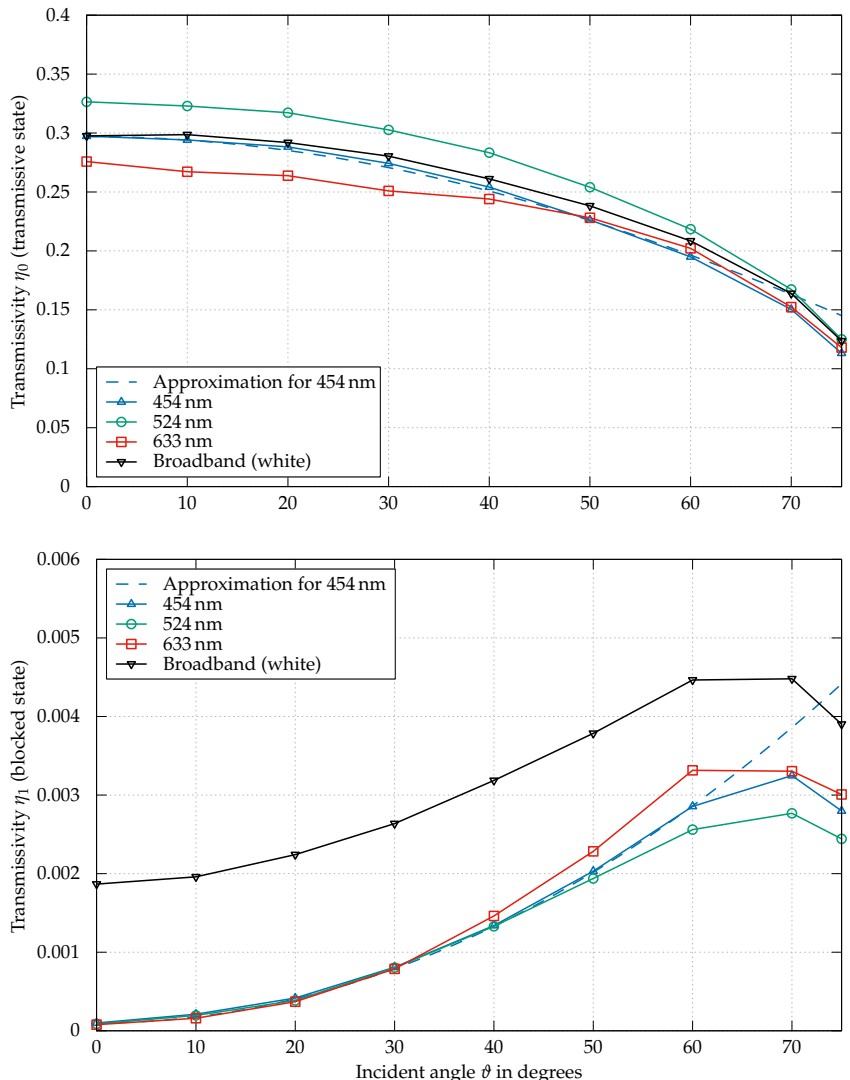

**Figure 2.** Transmissivity of the LC cell.

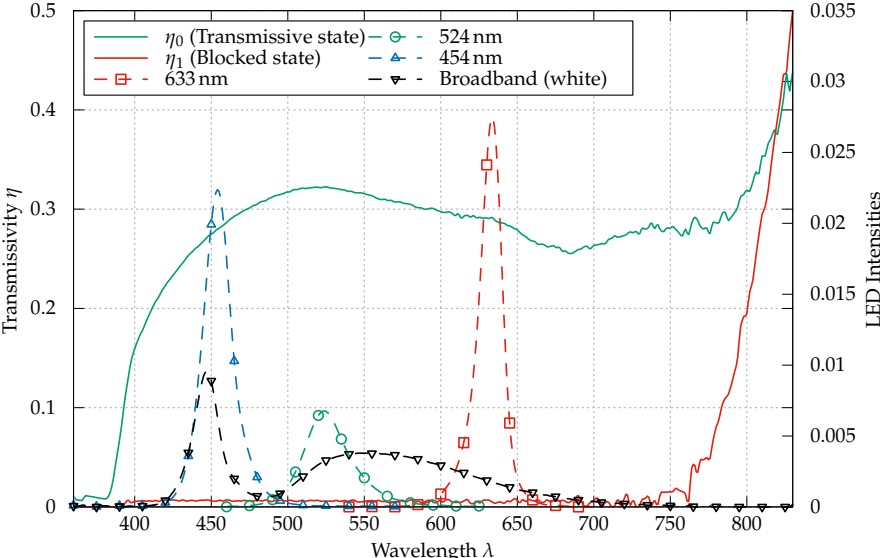

**Figure 3.** Spectral measurement results for the LC cell and the four different LEDs.

## 3. Interference Reduction

In the following, we distinguish two classes of interference and analyze the impact of LCD-based filtering in both cases by simulation. We assume a square LCD with an edge length of $l = 53.3\,\mathrm{mm}$ and $240 \times 240$ pixels. Likewise, the PD is square and of size $a = 3.6\,\mathrm{mm}$. With these assumptions and a focal length of $f = 15\,\mathrm{mm}$, thef ield-of-view (FOV) of the filter is approximately $120°$.

### 3.1. Ambient Light

The first type of interference under investigation is solar irradiation. Filtering can be accomplished in the electrical domain, e.g., by placing a high-pass filter after the first stage transimpedance amplifier (TIA). Nevertheless, interference by sunlight is critical since the detected and later on filtered interference power $P_i$ induces receiver side noise. Namely, shot-noise [3] in the PD with variance

$$\sigma_{\mathrm{shot}}^2 = 2qRP_iB. \tag{9}$$

Since the responsivity $R$ of the PD, the electron charge $q$ and the bandwidth $B$ are constant for a given receiver, the shot-noise is directly proportional to the interference power $P_i$. The relevance of this effect becomes apparent when comparing the potential high solar irradiation power with the signal power $P_s$. In such a case, $P_i \gg P_s$, the communication system is shot-noise limited with [3]:

$$\mathrm{SNR} = \frac{(RP_s)^2}{\sigma_{\mathrm{shot}}^2}. \tag{10}$$

One additional problem is receiver saturation by a large DC bias exceeding the dynamic range of the receiver.

In order to derive the potential of LCD-based filtering, we assume the sunlight to be purely diffuse and to arrive at the receiver from each direction with equal intensity.

By integration over a sphere with

$$\mathrm{d}A = \sin\vartheta\,\mathrm{d}\vartheta\,\mathrm{d}\varphi, \tag{11}$$

we obtain

$$F_{\mathrm{norm}} = \int_{-\pi/3}^{\pi/3} \int_{\pi/6}^{5\pi/6} \mathrm{d}A \tag{12}$$

as reference. This value represents the solar irradiation without an LCD in the optical path and with only a static aperture with an FOV of $120°$ remaining. For the interference, we can distinguish two portions, as shown in Figure 4. The first part

$$F_1 = \int_{-\pi/3}^{\pi/3} \int_{\pi/6}^{5\pi/6} \eta_1\left(\Psi\right) \mathrm{d}A \tag{13}$$

is interference leaking to the PD owing to $\eta_1\left(\vartheta\right) > 0$. The angle to the normal of the PD is given with:

$$\Psi = \arccos\left(\sin\vartheta\cos\varphi\right). \tag{14}$$

The second part

$$F_0 = \int_{-\Theta(0)/2}^{\Theta(0)/2} \int_{\Psi+\pi/2-\Theta(\Psi)/2}^{\Psi+\pi/2+\Theta(\Psi)/2} \eta_0\left(\Psi\right) \mathrm{d}A \tag{15}$$

is interference arriving at the PD by passing the LCD at the pixel positions opened for receiving the desired signal. By relating the remaining interference (13) and (15) to the case of not using a filter (12), we obtain the results on the SIR improvement by using the LCD-based filter approach:

$$\Delta \text{SIR} \left( \Psi \right) = \frac{\eta_0 \left( \Psi \right) F_{\text{norm}}}{F_0 + F_1}. \tag{16}$$

The factor $\eta_0 \left( \Psi \right)$ in (16) stands for the attenuation of the wanted signal by the LCD filter in transmissive state. Analytically derived results are presented in Figure 5, with the interference normalized with $F_{\text{norm}}$.

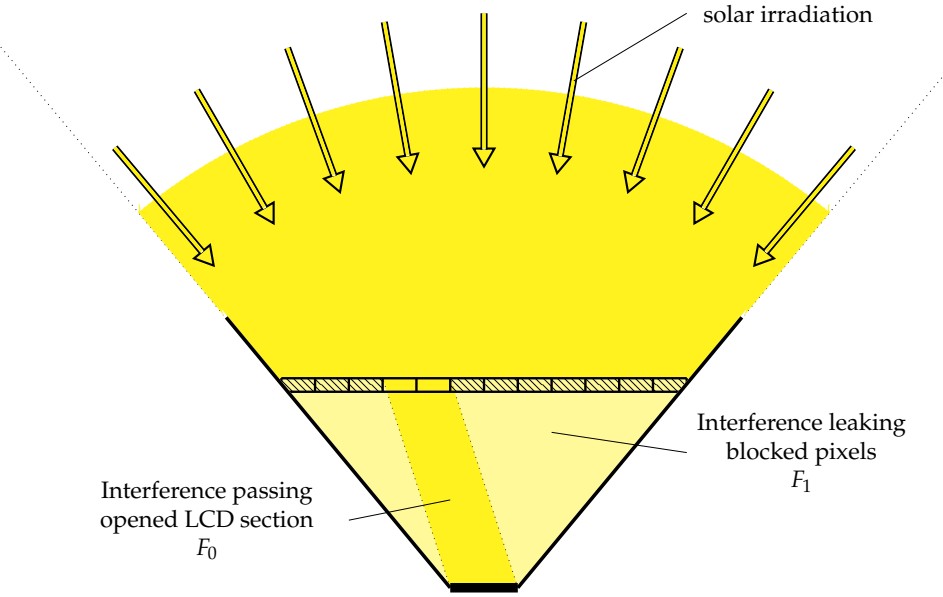

**Figure 4.** Reduction of solar-induced interference using the LCD filter.

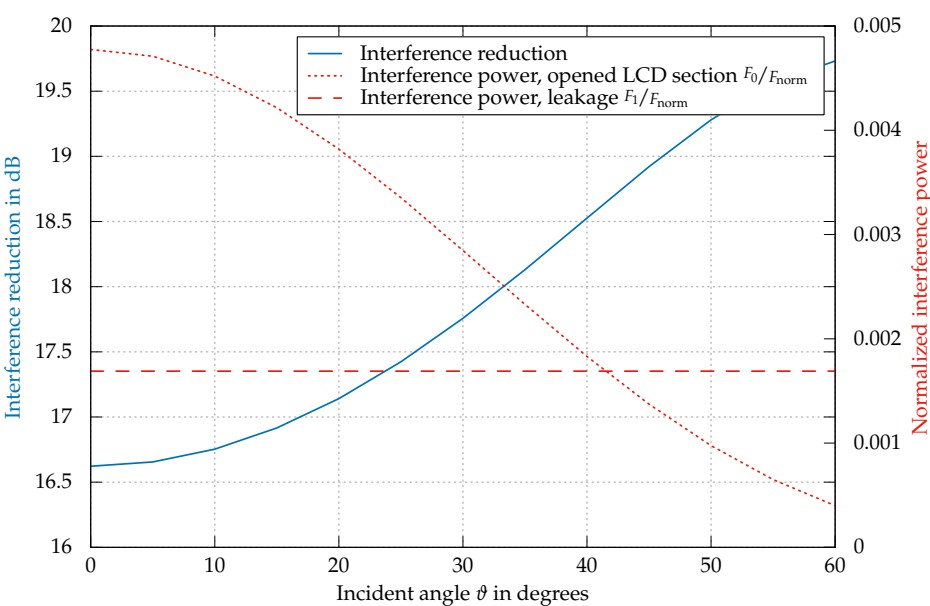

**Figure 5.** Suppression of unmodulated and diffuse interfering light sources using the method described in Section 3.1.

## 3.2. Modulated Interference

In a second scenario, we assume a point light source inducing interference at the receiver from a distinct direction. If such a light source is of unmodulated nature or only contains power-line frequency components [12], electrical domain filtering is possible. As already mentioned in Section 3.1,

the problem of receiver saturation remains. Typically, the power of such sources is in the same scale as the useful signal; thus, noise enhancement due to the interfering signal is not of relevance in general.

For modulated signals, DC-blocking is not a valid strategy. One possibility is an algorithmic approach, applying techniques like SIC, but this is limited to specific scenarios and increases the computational complexity at the receiver. Another example for modulated interference is multipath where delayed signal components deteriorate the receiver performance if not taken into account in the signal design or by algorithmic mitigation. Since this kind of interference is caused by reflections on the propagation path, we can assume the incoming directions to differ from the line-of-sight (LOS) path signal. Thus, direction-based filtering using an LCD is a very promising method.

In the following, we expect the interfering signals to be spatially separated from the desired signals by the minimum required angular difference for perfect interference suppression, derived in (5). In other words, we expect the interference signals to be attenuated by $\eta_1(\vartheta)$ while the useful signals pass the LCD with $\eta_0(\vartheta)$. This assumption leads to the results shown in Figure 6 with an interference reduction in the range of 18 dB to 35 dB for the blue-colored 454 nm LED.

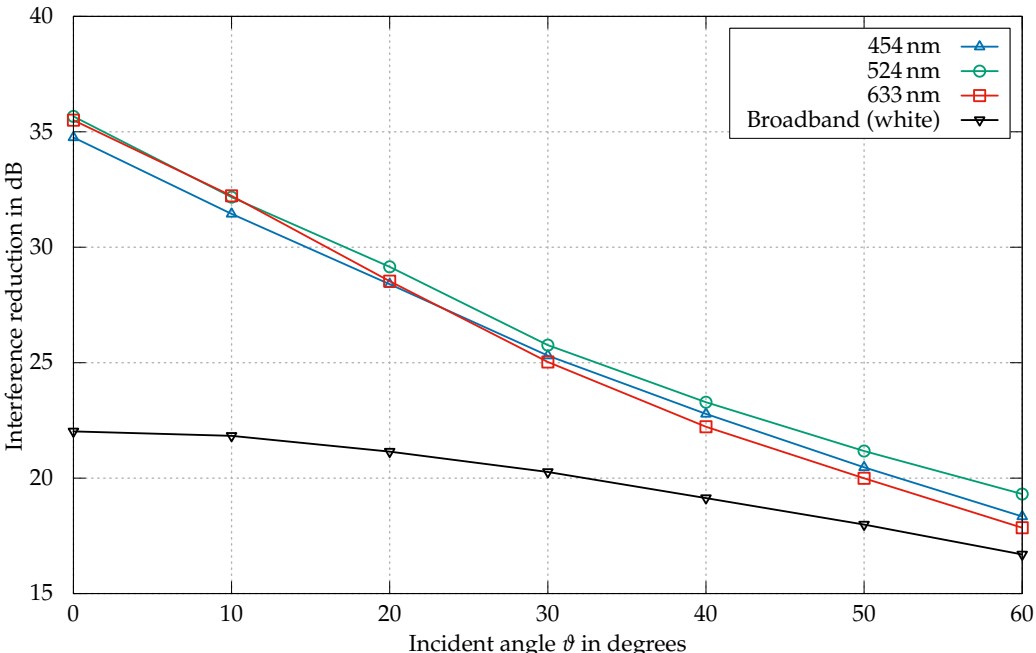

**Figure 6.** Reduction of modulated interference, according to the analytical method described in Section 3.2.

## 4. MIMO Communication

The numerical results reported so far are valid for the previously given application examples but are not directly applicable to MIMO and multi-user scenarios where the detection of multiple data streams is intended and receiver-side decorrelation methods are used. In order to compare our novel filter approach to a conventional MIMO receiver, we use the indoor scenario from [13] as reference. This is a $N_t \times N_r$ MIMO set-up with $N_t = 4$ LED transmitters mounted at the ceiling at a height of 2.5 m pointing downwards to the receivers located on a desk of 0.75 m height. The four transmitters with a FOV of 15° each are arranged in square configuration with a variable distance $d_t$ of 0.2 m, 0.4 m, and 0.6 m. The $N_r$ receivers are also arranged rectangularly and $d_r = 0.1$ m apart, enabling a reasonably small sized receiver ensemble.

Given this scenario, we extend the receivers by the novel LCD filter. Instead of four times replicating the receiver structure shown in Figure 1, a single LCD is used for simultaneously decorrelating all four PDs. Hereby, the number of required filter elements is reduced to one. This promising alternative is depicted in Figure 7. Both possible receiver structures will show equal

performance as long as the previously established requirements on the geometry are considered. One benefit of LCD-based filtering is a reduced receiver size since the required distance $d_{\mathrm{PD}}$ of the PDs is considerably smaller compared to the conventional MIMO system that relies on a sufficient large PD distance $d_r$.

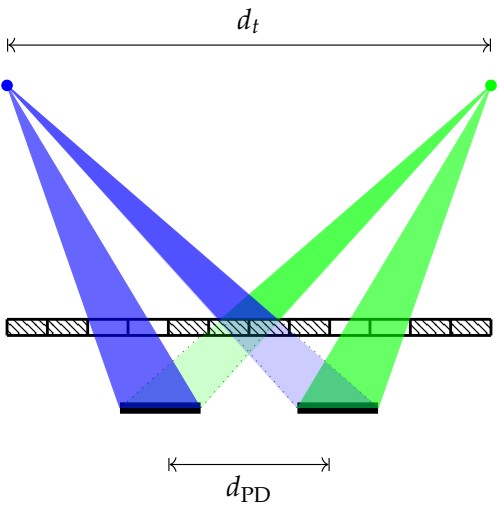

**Figure 7.** MIMO set-up using a single LCD for decorrelating two links.

In accordance with the conventions in [13], we have calculated the following channel matrices by applying the Lambertian law to the LOS paths:

$$
\mathbf{H}_{d_t=0.2\,\mathrm{m}} \approx 10^{-4} \cdot
\begin{bmatrix}
0.317843 & 0.000120 & 0.000120 & 0.000126 \\
0.000120 & 0.317843 & 0.000126 & 0.000120 \\
0.000120 & 0.000126 & 0.317843 & 0.000120 \\
0.000126 & 0.000120 & 0.000120 & 0.317843
\end{bmatrix},
$$

$$
\mathbf{H}_{d_t=0.4\,\mathrm{m}} \approx 10^{-4} \cdot
\begin{bmatrix}
0.272697 & 0.000134 & 0.000134 & 0.000138 \\
0.000134 & 0.272697 & 0.000138 & 0.000134 \\
0.000134 & 0.000138 & 0.272697 & 0.000134 \\
0.000138 & 0.000134 & 0.000134 & 0.272697
\end{bmatrix},
$$

$$
\mathbf{H}_{d_t=0.6\,\mathrm{m}} \approx 10^{-4} \cdot
\begin{bmatrix}
0.201909 & 0.000137 & 0.000137 & 0.000000 \\
0.000137 & 0.201909 & 0.000000 & 0.000137 \\
0.000137 & 0.000000 & 0.201909 & 0.000137 \\
0.000000 & 0.000137 & 0.000137 & 0.201909
\end{bmatrix}.
$$

Since no analytical solution for calculating the MIMO intensity modulation with direct detection (IM/DD) capacity is known, modulation constrained MIMO performance figures are derived in [13]. Perfect channel knowledge, perfect time synchronisation and receiver-side maximum likelihood (ML) detection

$$
\hat{\mathbf{u}} = \arg\max_{\mathbf{u}} p_{\mathbf{y}}\left(\mathbf{y}|\mathbf{u},\mathbf{H}\right) = \arg\min_{\mathbf{u}} \|\mathbf{y} - \mathbf{H}\mathbf{u}\|_{\mathrm{F}}^2 \tag{17}
$$

are assumed for estimating the transmit vector $\mathbf{u}$. Lower bounds for the BER of spatial multiplexing (SMP) and spatial modulation (SM) are given in [13], closely approximating the ML decoding results for BERs $\leq 10^{-2}$. In the case of SMP, independent data streams are simultaneously transmitted. Each one is modulated using unipolar $M$-ary pulse amplitude modulation ($M$-PAM). SMP is a parallel scheme offering a constrained capacity of $N_t \cdot \log_2(M)$ bits/s/Hz with the transmit power equally distributed on all $N_t$ transmitters. The use of $M$-PAM is reasonable owing to its good optical power

efficiency, e.g., shown in [14]. SM additionally exploits the spatial domain by assigning individual bit patterns to the transmitters. With one transmitter active at a time encoding the source bits in the spatial domain and as conventional signal constellation, a transmit rate of $\log_2(N_t) + \log_2(M)$ can be achieved. Since a zero power constellation point would not allow the receiver to estimate the spatial information, the $M$-PAM constellation is altered in the case of SM to avoid the zero symbol. This modification reduces the minimum distance compared to conventional $M$-PAM but enables the use of the spatial domain.

Results on the upper bounded BER at fixed efficiencies of 4 bits/s/Hz and 8 bits/s/Hz are shown in Figure 8. Generally speaking, the LCD filter offers a strong decorrelation of the signal sources at the cost of a decrease in signal power and by this a relative increase in receiver-side noise. For SM and SMP, the decorrelation gain by using the filter clearly outweighs this signal power reduction. Compared to the conventional MIMO receiver with a transmitter distance of $d_t = 0.4$ m, the SNR gain is approximately 11 dB for SM and 27 dB for SMP. With a lower distance of $d_t = 0.2$ m, these gains increase to 18 dB for SM and 40 dB for SMP. The higher gains for SMP show that the impairment of the modified constellation diagram for SM is outbalanced by the additional degree of freedom. At low transmitter distances, the conventional receiver performance deteriorates because of decreasing channel decorrelation. In the case of $d_t = 0.6$ m, the assumption of a very narrow receiver-side FOV in [13] leads to a perfect separation of the different sources without any filtering necessary. This result is questionable since diffuse signal components introduced by reflections are not taken into account and will destroy the assumed ideal scenario. The similar applies to a change of position/orientation of transmitter/receiver.

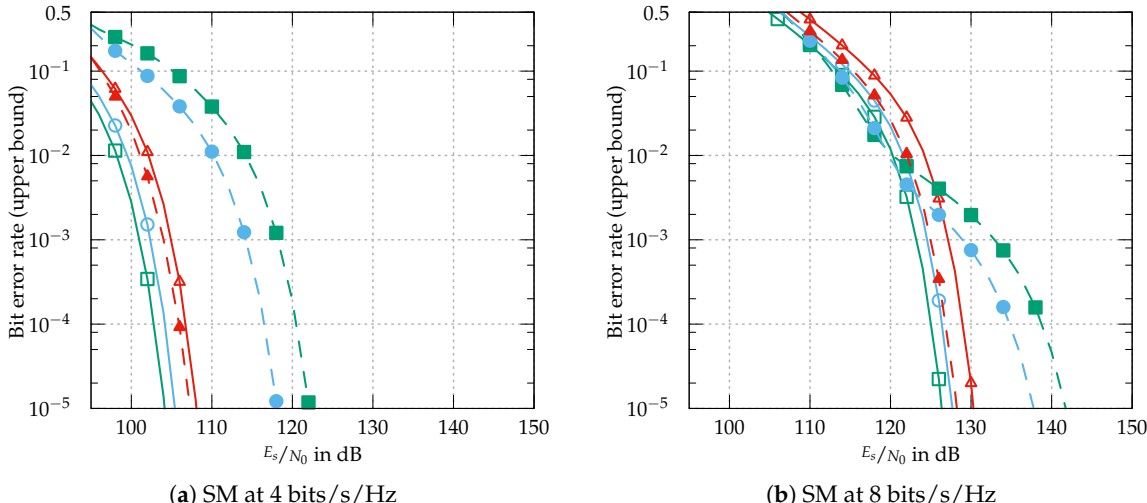

(**a**) SM at 4 bits/s/Hz

(**b**) SM at 8 bits/s/Hz

**Figure 8.** *Cont.*

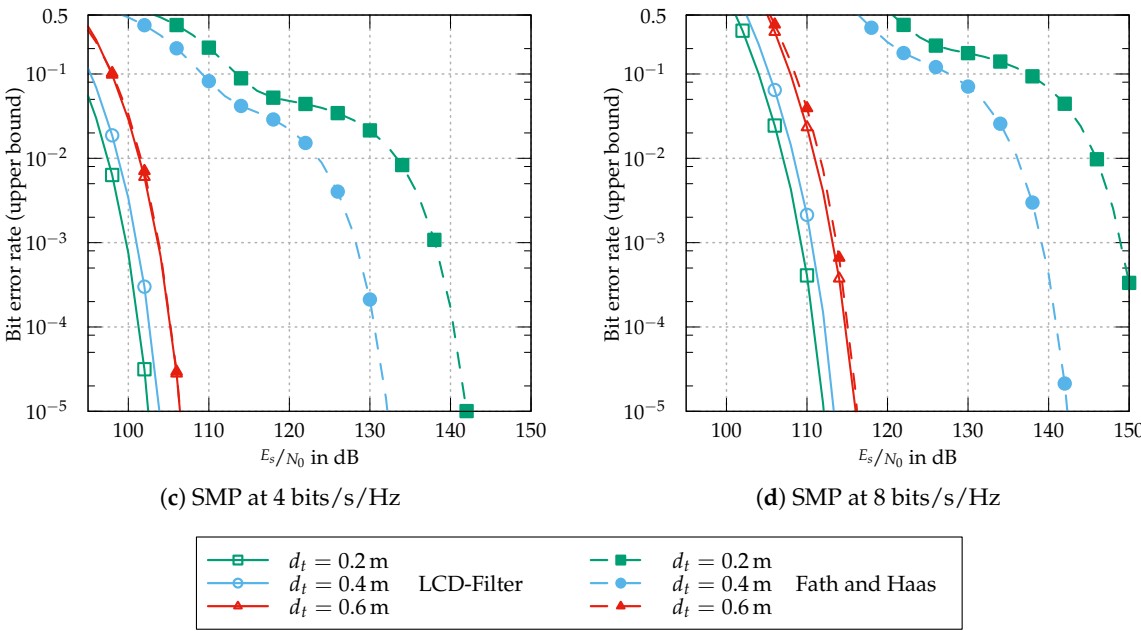

**Figure 8.** Comparison of LCD filter performance to conventional MIMO results presented in [13].

## 5. Conclusions

In this work, we introduce a novel method for optical receive filtering that allows for actively steering the receive pattern of an optical detector. Accordingly, it is possible to utilize the angular information of the incoming signals as filter criteria leading to increased receiver performance in comparison with conventional filtering methods. The measurement and simulation results presented in this work support the practical relevance of the proposed interference suppression method. Thus, interference suppression is not restricted to masking other modulated sources but includes, among others, the reduction of interference-induced noise and the mitigation of multipath distortions. One further application is the suppression of self-interference in full-duplex communication, e.g., when the transmit signal of an optical underwater modem is reflected back by particles in the water column. Additionally, the possibility for MIMO decorrelation using an LCD filter is presented.

As a by-product the proposed filter can further be used to extend the dynamic range of a given optical receiver by attenuating the optical signal, e.g., enabling the use of a sensitive detector in a short-range communication scenario.

Regarding filter performance, the proposed direction-based filtering method implements a filter criteria independent from wavelength-based methods. Thus, enhancing the overall performance by combining both methods is possible. A comparison of the dynamic LCD-based filter to static approaches like imaging receivers strongly depends on the chosen parameters. Generally speaking, the static methods can offer superior performance while being limited to specific scenarios. Moreover, positioning and orientation is critical. The novel LCD filter, on the other hand, can be dynamically adapted. Thus, conventional angle-diversity filters can be beneficially applied in well-defined scenarios like free-space point-to-point links while the adaptive LCD filter approach shows advantages in mobile applications, for example. Unlike the state-of-the-art angle-diversity receivers, the LCD-based filter can be used for interference suppression using a single PD while the competing decorrelation methods require at least two PD receivers.

A challenge is that the LC properties limit the optical performance. With the ongoing demand for energy-efficient and high-contrast ratio displays, especially for the use in mobile devices, further improvements of the LCD technology can be expected. This is supposed to lead to a further increase of the key performance parameters for optical filtering using the proposed method. However, the novel adaptive filtering is not limited to the LCD-based approach and is directly applicable to

alternative display technologies like MEMS-based digital microshutter displays [15] and transmissive electrophoretic [16] displays as well.

**Author Contributions:** Conceptualization, G.J.M.F. and A.K.; Writing—original draft, G.J.M.F.; Writing—review and editing, A.K. and P.A.H.; Supervision, P.A.H.

**Funding:** This research received no external funding.

**Conflicts of Interest:** The authors declare no conflict of interest.

## Abbreviations

The following abbreviations are used in this manuscript:

| | |
|---|---|
| BER | bit error rate |
| FOV | field-of-view |
| IM/DD | intensity modulation with direct detection |
| LC | liquid-crystal |
| LCD | liquid-crystal display |
| LED | light emitting diodes |
| LOS | line-of-sight |
| ML | maximum likelihood |
| MIMO | multiple-input multiple-output |
| PAM | pulse amplitude modulation |
| PD | photodiode |
| SIC | successive interference cancellation |
| SIR | signal-to-interference ratio |
| SM | spatial modulation |
| SMP | spatial multiplexing |
| SMU | source-measurement unit |
| TIA | transimpedance amplifier |
| VLC | visible light communication |

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
