# Peer review of "Optical Interference Suppression Based on LCD-Filtering"

_applsci, doi:10.3390/app9153134_

Round 1

Reviewer 1 Report

The authors propose a filtering method based on a liquid-crystal display (LCD) to reduce the effect of interferences in optical communication. The method is new and interesting and has potential for applications where modulated noise is a problem.

The paper however needs some improvement and further discussion to be considered for publication.

Several points present assertions which are neither obvious nor proved, making it difficult to evaluate the robustness of the results. Generally, it is difficult to evaluate the advantages and limitations of the method compared to other solutions. Also missing is an experimental or empirical validation of the method to show the claimed performance. These points (advantages and disadvantages, applicability and performance compared to benchmark solutions) should be included in the paper.

Here some general issues which should be addressed:

1) In the beginning, the authors claim: «The proposed liquid-crystal display (LCD)-based filtering method can be favorably applied in optical underwater communication systems to isolate weak receive signals from high power interference and by this enable the use of sensitive avalanche photodiodes».

In the paper, however, there is no mention of underwater communication and avalanche photodiodes. This sentence is not supported in the text and should be discussed in the introduction and conclusions. Also, the experimental setup does not include an APD.

2) The introduction is not exhaustive. State of the art should be improved to include the different methods and schemes used for optical interference suppressions in optical communications.

3) The conclusions should present a broader comparison with other solutions (benchmarking). In which cases should the proposed method be used and in which ones should it not? The described method is advantageous if the interfering light has a specific direction, which is well distinct from the desired signal. This is a strong limitation of the applicability. A discussion should be included.

4) The formalism in the equations is not always consistent and according to standard conventions. Angles should be in radians and not in degrees (equations 6, 7, 10). The surface element is not a vector and should not be indicated as such (equations 10 and 12). The integrals in equation 10 should not include the variables in the extremes and again not in degrees.

5) English is not always clear. Sentences are very long and sometimes unclearly structured. The usage of commas is also not always correct.

6) A list of abbreviations at the end of the paper could be helpful for the reader.

The following specific issues should also be addressed to improve the clarity of the paper.

7) Page 2, Line 41-43: The authors claim: ” Different filter types like colored glass filters and dichroic filters are available, sharing this principle dilemma.» Which dilemma? This is unclear.

Also, the authors write: «angular derivation of the filter response». What does it mean? Do the authors mean that the filtering properties depend on the angle of incidence? Actually, this is true only for interference filters. Color glass filters are insensitive since based on absorption.

8) Page 2, line 59: Insert a space instead of the point in “almost.perfect”.

9) Page 3, equation 1: Equation 1 holds only if a<<l. This should be mentioned in the text.

10) Page 3, bottom: The type and manufacturer of LC-Cell should be included.

11) Page 3, bottom: The description of how the spectra of figure 3 were acquired is missing. Which instrument was used?

12) Page 5, lines 99-101: Why these dimensions? This should be clarified. The PD used before is a 10x10mm2. Now, a^2 is 3.6x3.6 mm2.

13) Page 5, equation 8 and 9: A reference to explain these equations should be added since these are not derived in the paper.

14) Page 6, figure 2: “degrees” instead of “degree”

15) Page 7, figure 3: Numbers and units of the right y-axis are missing.

16) Page 7, lines 108-111. The authors claim: “For the interference remaining we can distinguish two portions. As shown in Figure 4, the LCD is leaking some of the irradiated power to the PD due to eta(theta)>0. The second part is interference that arrives at the PD by passing the LCD at the pixel positions opened for receiving the useful signal.» This sentence is not clear. The two regions should be marked in fig. 4.

17) Page 8, figures 4 and 5: “degrees” instead of “degree”. Both figures show results from calculations and not actual measurements. This should be explicitly written in the caption of each figure.

18) Page 8, figures 4 and 5: Both figures show simulation results. This should be explicitly written in the caption of each figure.

19) Page 8, figure 4: Numbers on the right y-axis are missing.

20) Page 8, figure 4: How is the interference power normalized? This should be explained in the text.

21) Page 9, section 3.3: The comparison of the filter performance is done using the method of reference [11]. The motivation for this, as well as a minimal explanation of the method, should be included.

Author Response

Dear Reviewer,

thanks for your very detailed response and helpful comments. Regarding your general suggesting for improvement I have added a paragraph on competitive filtering methods in the conclusions. Similar applies to the introduction where the alternative methods are described in more detail now. Moreover Section 3.1 is heavily revised to improve the comprehensibility. Most other changes follow your suggestions 1) - 21). The Section on MIMO Communication is extended owing to remarks of a second reviewer

To your suggested improvements:

1) Removed the "Featured Application" section

2) Extended the paragraph on the state-of-the-art methods

3) Added a paragraph regarding benchmarking in the conclusions. Your second statement is not completly true in my opinion since all angular-diversity receivers face this challenge when decorrelating light sources. The advange of the LCD-based filter is that non-directed interfernce can be filtered (compare Section 3.1) (using a single PD).

4) Corrected

5) Improved on this

6) Added list of abbreviations

7) Reordered sentences and be more specific now

8) I do not have this error in my version of the manuscript, strange

9) Correct by adding this requirement

10) The manufacturer is not known. We purchased an very cheap (~3$) welding helmet and extracted the LC cell

11) Added paragraph on spectrometer and interpretation of LC cell wavelength dependency

12) The Thorlabs FDS100 exhibits an active area of 3.6mm x 3.6mm, the edge length a=3.6mm is now specified instead of the detector are a^2 to make things more clear

13) Added refrence directly to the equations to state the source more clearly

14) Corrected, same applies to Figure 4 and Figure 5

15) Figure 3 now displays the measured values instead of the normalized ones. The actual values are not of great importance in the further discussion since the transmissivity is a multiplicative factor

16) Restructured and extended the explanation. A new Figure 4 is introduced displaying the different sources of interference

17) Corrected

18) Added remark in figure captions

19/20) Fixed

21) Added explanation

Again, thanks for this very nice review.

Reviewer 2 Report

Forkel et al. report about the use of liquid-crystal displays to suppress optical interference in visible light communication application. The work is of interest for Applied Sciences readers, the introduction is exhaustive and the research design appropriate, but the description of methods and presentation of results should be improved.

1) As a general comment, the authors talk about measurements and simulation (for example in the conclusion, rows 154-155 “The measurement and simulation results presented in this work support the practical relevance of the proposed interference suppression method”) , but the main results of the work arise only from simulation. This aspect should be more clear.

2) Please, check the dimension of equation (9).

3) Methodology is described only superficially, especially in par 3.3 MIMO Communication: even if the methods are already reported in ref 11, some details about the model should be added to help the reader (maybe in a Supplementary Material?)

Author Response

Dear Reviewer,

thanks for your review.

To your suggested improvements:

1) Marked simulation results more clearly in the caption of the figures

2) The equation was taken from [3] what is stated more clearly now. For the units I can not see a problem: (R^2 * P^2) / (q * R * P * B) = (A^2/W^2 * W^2) / (A*s * A/W * W * 1/s) = A^2 / A^2

3) Tried to improve on this:

        - Added paragraph on spectral measurement method

        - Updated multiple Figures (and text) to clarify normaized values

        - Equations in Section 3.1 on solar irradiation are explained in more detail now and a new Figure 4 is added for explanation

        - Section 4 on MIMO Communication was heavily revised, according to your suggestion. This includes a new Figure 7 and a more detailed explanation on the MIMO scenario

best regards

Round 2

Reviewer 2 Report

The authors addressed all the points and the manuscript is acceptable in the present form as is.